# The Effect of FIFA 11+ on the Isometric Strength and Running Ability of Young Soccer Players

**DOI:** 10.3390/ijerph192013186

**Published:** 2022-10-13

**Authors:** Xin Zhou, Anmin Luo, Yifei Wang, Qingqing Zhang, Yu Zha, Sicheng Wang, Caroline Ashton, John Ethan Andamasaris, Henry Wang, Qirong Wang

**Affiliations:** 1School of Rehabilitation Medicine and Health, Hunan University of Medicine, Huaihua 418000, China; 2National Institute of Sports Medicine, National Testing & Research Center for Sports Nutrition, 1 Anding Road, Chaoyang District, Beijing 100029, China; 3School of Sports Medicine and Health, Chengdu Sport University, Chengdu 610041, China; 4Beijing Haidian Experimental High School, Beijing 100089, China; 5School of Kinesiology, Shanghai University of Sport, Shanghai 200438, China; 6School of Kinesiology, Ball State University, Muncie, IN 47306, USA

**Keywords:** FIFA 11+, youth soccer, H/Q, performance, injury prevention

## Abstract

Soccer is the world’s game, and keeping athletes healthy while playing the game has often been a focus of study. There is a high occurrence of musculoskeletal injuries reported in soccer. FIFA 11+ was developed as an intervention to help prevent such injuries. FIFA 11+ has previously been studied for its efficacy as an injury prevention program, but not for its effect on sports performance in an adolescent population. The purpose of this study was to look at the effect of implementing the FIFA 11+ intervention on strength, speed, and agility. Twenty youth soccer players were randomly divided into an intervention group (INT) and a control group (CON). The intervention lasted for eight weeks and performance assessments were completed pre- and post-intervention. Post-test INT knee flexor strength was significantly higher than pre-test scores (*p* < 0.05). INT also demonstrated significantly higher hamstring to quadriceps strength ratio (H/Q) after the intervention (*p* < 0.05), while the CON H/Q did not change significantly. 30-m sprint performance of both groups improved from pre- to post-test (*p* < 0.05). Shuttle run performance was significantly improved in post-test scores for INT players (*p* < 0.05), but did not change significantly for the CON players. It is suggested that implementing FIFA 11+ before training in young soccer players can lead to performance benefits as well as injury prevention benefits.

## 1. Introduction

Soccer is the most popular sport in the world among people in different countries [1], but it often presents a high risk of injuries for musculoskeletal damage [2]. A statistical analysis of sport injuries over a 10 year period recorded 19,530 total injuries in 17,937 recreationally active participants [3]. 35% of those occurred during participation in soccer, making it the activity with the most injuries [3]. Elite soccer players suffer between 1.5 and 7.6 injuries per 1000 h of training and between 12 and 35 injuries per 1000 h of matches [4]. In youth soccer, the incidence of injuries during matches has been shown to increase with age from 1 to 5 injuries per 1000 match hours for players aged 13–15 years, to 15 to 20 injuries in the same period for players aged over 15 years [5]. These injuries can often cause loss of playing time for athletes, and loss of economic resources for clubs [6]. In response to high injury rates, the FIFA Medical Assessment and Research Center (F-MARC) developed FIFA 11+, a comprehensive warm-up program designed to prevent musculoskeletal injury. FIFA 11+ is broken down into three sections: low to moderate intensity running for eight minutes; strength, plyometric, and balance exercises for ten minutes; and high intensity running exercises for two minutes. The purpose of this intervention is to induce beneficial neuromuscular effects by improving conscious control over running, sudden cutting, jumping, and landing [7]. Previous research has supported that the FIFA 11+ intervention reduces incidence of injuries for players at all levels [7].

For soccer players, strength is not only an important part of physical fitness, but also a factor related to injury risks [8]. The relationship between isometric strength and injury occurrences has been widely discussed, and is often used to predict and monitor injuries of athletes [9]. High-level soccer players sprint 11% of the time in a game [10]. Previous research has proved that athletes with higher running velocities have a relatively lower risk of injuries [11]. Thus, sprint abilities not only can be used to determine the outcome of a soccer game but also can be regarded as a regulatory factor of injury risk as well. Researchers often associate and quantify injury risk with the ratio of concentric hamstring strength to concentric quadriceps strength [12]. The ability of the hamstring to stabilize the knee joint during dynamic movement reduces injury risk, which indicates that a larger hamstring-to-quadriceps-strength ratio can be beneficial for injury prevention [12]. Compared with elite players, unbalanced knee joint strength is more common in adolescent players who have relatively low muscular strength when in a stage of growth and development [13]. The relative weakness and immature movement patterns put adolescent athletes at a higher risk of injuries [13].

FIFA 11+ has also demonstrated performance benefits when the intervention has been proven to improve speed and agility in soccer players [14]. The study which reported significant improvements in vertical jump height and sprint speed only investigated the effectiveness of the intervention program on adults [14]. Further research is still needed to determine the efficacy of FIFA 11+ on sports performance on a youth population. The purpose of this study therefore was to evaluate the effectiveness of the FIFA 11+ intervention on performance components in adolescent soccer players compared to a typical warm-up. Performance components of interest included isometric knee strength, speed, and agility. Research has shown performance benefits in adult participants and it was expected that similar benefits would appear in adolescent participants [12,15,16]. It was hypothesized that the FIFA 11+ program would help improve knee muscular strength, as well as speed and agility.

## 2. Methods

### 2.1. Participants

Twenty male soccer players in the U14 age group were selected to participate in this study. The participants regularly trained five times a week for about 120 min each session, including three general training sessions, one video analysis session, and one team match per week. All participants had been in regular training for the last three years. Participants were all healthy with normal development and had no muscular injuries that would affect training performance. Only field players were included in the study.

### 2.2. Experimental Procedures

The 20 subjects were randomly divided into two groups by a random number table, the FIFA 11+ intervention group (INT) and the control group (CON), with 10 subjects in each. During the course of the experiment, the INT players completed the FIFA 11+ exercises, while CON players completed their regular warm-up before training. Each warm-up lasted 20 min. The study lasted eight weeks and interventions were conducted three times per week, prior to their regularly scheduled general training sessions. All of the players maintained normal routine, training, learning, and playing throughout the study. Players did not perform additional strength training outside of normal training time. The team members were responsible for recording the attendance of each player and using GPS (Catapult Open Field) to monitor the running distance and duration of each training session.

### 2.3. FIFA 11+ Intervention Plan

The FIFA 11+ warm-up consisted of three parts. The first part included low to moderate intensity running and dynamic stretching for approximately eight minutes. The second part then consisted of exercises aimed at developing strength, balance, muscular control, and core stability for 10 min. The last part was a combination of higher intensity running and soccer specific drills for two minutes [17].

### 2.4. Control Warm-Up Program

The control group players completed typical warm-up activities that they were previously accustomed to performing before training sessions.

Running at the pace of about 6 min/km for 3 min. Stretching for 4 min in total, half kneeling hamstring stretch; side to side lunge with reach; dynamic squat stretch; standing calf and hamstring stretch, with each stretching action lasting for 20 s each time for 3 times. Running for 3 min, which includes a sprint of 30 m for 1 min, 10 m of change of direction speed training for 2 min. 10 min of ball practice: ball passing for 5 min, ball dribbling alternately for 5 min (20 m distance). The control warm-up was completed during the same 20-min time span as the intervention warm-up.

### 2.5. Muscular Strength Test Method

Strength tests were all performed by the same therapist with two years of experience using a handheld dynamometer (MicroFET3, Hoggan, Salt Lake City, UT, USA). Testing locations were performed consistently according to dynamometer manufacturer instruction. Before the test, all participants performed a five-minute warm-up, including jogging and dynamic stretching. Testing protocols were demonstrated to the players, and main points and precautions were explained prior to testing. A warm-up trial was conducted to familiarize the subjects with the test involving a 50% effort held for five seconds. Each muscle group was then tested three times, with a rest period of one minute between trials. The average for the trials was calculated. Muscle groups tested included knee flexors, and knee extensors. For knee extensor testing, the participants sat in a chair with hips and knees at 90 degrees. The handheld dynamometer was applied to the anterior lower leg just proximal to the ankle joint. For the knee flexor testing, the participants sat in the same position except the dynamometer was applied posterior lower leg, just superior to the calcaneus. Participants were instructed to push or pull against the dynamometer with maximum effort without moving their upper leg. Absolute strength was represented as maximum voluntary contraction under isometric conditions (MVC). Relative strength was calculated by dividing MVC by the subject’s body weight (BW) and used for comparisons between subjects. Knee strength balance was also quantified by dividing hamstring strength by quadriceps strength (H/Q).

### 2.6. 30 m Sprint Test Method

The players performed three separate 30 m sprint tests with 4-min intervals [18]. The segmented timing system was used to record the time (Smart speed, Australia, Fusion Sport) and the fastest time record was used as the final result.

### 2.7. Shuttle Runs Test Method

5 training cones were placed on the soccer field at the distance of 5, 10, 15, 20, and 25 m respectively. In the beginning of the test, players would run and knock down the 5 m training cone, and then run back to the starting point to knock down the training cone at the starting line (the training cone at the starting point is required to be put back every time it is knocked down). The players proceeded to knock down the 10 m training cone, and the cycle would continue until all the training cones were down at the end of the test [19]. The test would be performed once by the players due to potential fatigue caused by the drills. The split timing system was used to record the time (Smart speed, Australia, Fusion Sport).

### 2.8. Statistical Analysis

Data and statistics in the study were analyzed by SPSS20.0. For continuous variables that conformed to normal distribution, comparison of results between groups was conducted using Independent Samples *t*-test. Comparison of data before and after within-group tests were conducted using Paired-Samples *t*-test. The results were expressed as mean ± standard deviation (±s). Strength was normalized by dividing MVC by BW. Ratio of hamstring to quadriceps strength (H/Q) was also calculated as a measure of relative knee strength balance.

## 3. Results

All 20 participants completed testing. There was no significant difference in height, weight, age, body mass index (BMI), or body fat percentage between the two groups before or after the experiment (Table 1). There was no significant difference in low-speed running, high-speed running, maximum speed, and training load between the two groups throughout the eight-week intervention (Table 1).

### 3.1. Muscular Strength Testing in Knee Joint

After eight weeks of FIFA 11+ there was a significant difference in knee flexor strength in INT players, with greater strength observed post-intervention (*p* < 0.05). The change in the CON players was not significantly different from pre- to post-tests. Knee extensor strength of the INT players was slightly reduced after FIFA 11+ (*p* < 0.05), but did not change significantly for the CON players (Table 2). As a result, the H/Q of maximum strength was significantly higher in post-testing for the INT players compared to CON players (*p* < 0.05) (Figure 1).

### 3.2. Running and Agility Tests

The 30-m sprint times of the INT players and the CON players were significantly better in post-testing than pre-testing (*p* < 0.05). The results of the inter-group comparison illustrated that the post-test 30-m sprint times of INT players were significantly better than those of the CON players (*p* < 0.05) (Table 3). The 5 × 25-m shuttle run time for the INT players was significantly better post-intervention than pre-intervention (*p* < 0.05), while the CON players had no significant change.

## 4. Discussion

This study illustrated the impact of FIFA 11+ on muscular strength and running performance of young soccer players. The results found that after eight weeks of the FIFA 11+ intervention, there was a significant improvement in knee flexor strength in the INT group (*p* < 0.05). FIFA 11+ has been widely proven to improve the isokinetic strength of the knee flexors of soccer players [20], but previous research did not prove whether it has the same effect on isometric strength. Our results suggest that the isometric strength of knee flexors has increased.

This is likely due to the strength training portion of the FIFA 11+ intervention, which includes Nordic hamstring training. It can produce specific neural adaptive changes that can enhance the effect of muscle activation and induce the increase of muscle motor unit recruitment, which is the main mechanism to achieve the increase of muscle strength in the short term [21]. The positive effect of this improvement is that it may reduce the risk of hamstring injury. Hamstring injuries are very common in soccer, and often occur by non-contact mechanisms. Reduced isometric strength of the knee flexors is considered a risk factor for hamstring injury [22,23,24].

H/Q of the INT group was also significantly higher following the FIFA 11+ intervention (*p* < 0.05). These results were similar to the results seen in previous studies on adult populations. Brito et al. observed that FIFA 11+ could improve the balance of bilateral knee flexion and extension strength of male high level soccer players [15]. Nawed et al. found that similar results were also observed in futsal players [14]. Impellizzeri et al. proved that performing part two of the FIFA 11+ intervention 2–3 times a week alone can improve the strength balance of the knee flexors and extensors of amateur soccer players [16]. Owing to the sports characteristics of soccer, the quadriceps need to bear weight during vertical jump take-offs and landings [13]. As a result, the quadriceps muscles often become stronger than hamstrings [13]. As soccer players tend to have lower H/Q than the average population [12], this strength imbalance is a key risk factor for knee injury [22]. The improvements demonstrated by FIFA 11+ on the adolescent population have significant implications for injury prevention since a higher H/Q ratio is associated with a lower risk of injury [25,26,27,28].

The second part of FIFA 11+ is an important cause for the improvement of knee strength balance [27,28,29,30,31,32,33]. Resistance training synergistically activates antagonistic muscle nerve pathways during muscle contraction [29], and as a consequence, the increase in active muscle strength may also be related to the decrease in antagonistic muscle activation [34].

Results of the current study found that the relative strength of the knee extensors of the INT players was significantly lower than pre-intervention. This result was not supported by similar studies on adults. While it did contribute to increasing H/Q ratio and is not considered a risk factor, decreases in knee extensor strength are certainly not the goal of the intervention.

The current study demonstrated that FIFA 11+ can enhance the running ability of young soccer players, especially in comparison to traditional warm-up exercises. The 30-m sprint and 5 × 25-m shuttle run are commonly used performance indicators for soccer players [35]. These tests reflect the explosive power, endurance, and neuromuscular control of the athletes’ lower limbs, and are widely used as fitness tests for soccer players of all age levels [35]. The results of this study are supported by similar findings in adult soccer athletes and older adolescent futsal players [14,36,37].

FIFA 11+ promotes performance improvements through two mechanisms, the first being the increase in muscular strength. Malone et al. determined that increasing maximum and relative strength may also improve athletic performance [38]. Therefore, the increase in the strength of knee flexors seen in the INT players may be a contributing factor for the increase in sprint speed. As the main mover of the horizontal component of the ground reaction force (GRF), the knee flexors are an important mechanical indicator of acceleration and sprint performance [39]. Knee flexor strength plays an important role in generating propulsion, and can offset the knee joint damage caused by GRF during sprinting [40].

Secondly, FIFA 11+ includes plyometric training, such as vertical jumping, box jumping, and bounding. Plyometric training has been proven to enhance lower limb muscular power, neuromuscular recruitment, and muscle coordination [41]. A recent meta-analysis suggests that lower-limb plyometric training has a small to moderate positive effect on jumping, sprint performance, and lower-limb muscular strength in healthy adults in amateur or professional sports [42]. In the long run, the simultaneous development of running ability and strength may help reduce the risk of player contact injury [42,43]. Faster sprint speed and the ability to run back and forth can act as a potential injury risk modifier and is often associated with better neuromuscular function and aerobic capacity [38].

## 5. Limitations

This study focused on the effects of FIFA 11+ on isometric strength of relevant muscle groups, and related running ability in young soccer players to further enrich the theoretical system of FIFA 11+. According to the purpose and design of this study, incidence of player injury was not reported, and there was no follow-up after the experiment. In order to clarify the effect in practice, it should be further investigated whether the beneficial neuromuscular performance improved by FIFA 11+ can reduce the risk of common injuries. There should also be more research to analyze the relationship between common injuries and isometric muscular strength in young soccer players.

## 6. Conclusions

This study concluded that FIFA 11+ can significantly enhance the isometric strength of the knee flexors in adolescent soccer players. It can improve the strength balance of the knee joint, and significantly improve performance in 30-m sprint and 5 × 25-m shuttle run tests. Further investigation into the reduction of knee extensor strength seen in youth athletes is still needed, and appropriate adaptations may need to be considered. Overall, it is suggested that implementing FIFA 11+ before formal training in young soccer players can lead to performance benefits in addition to injury prevention benefits.

## Figures and Tables

**Figure 1 ijerph-19-13186-f001:**
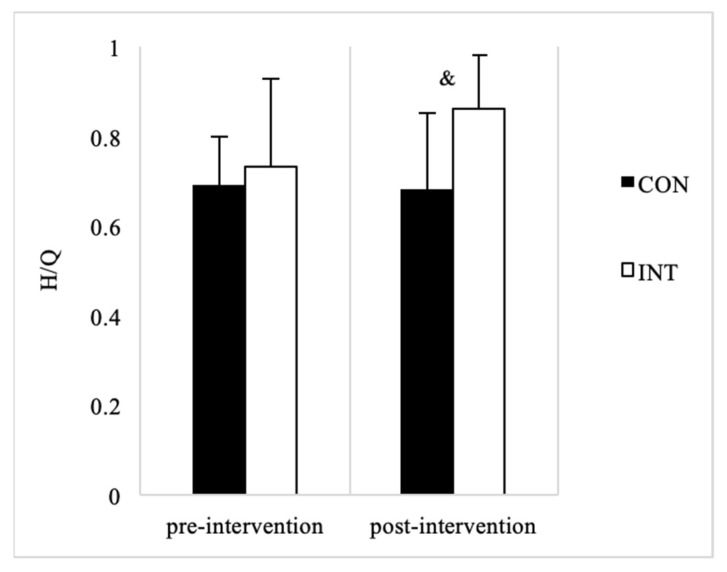
Knee Strength Ratio (H/Q). Note: Comparison between the control group and the intervention group during the same period: & = *p* < 0.05.

**Table 1 ijerph-19-13186-t001:** Baseline data.

Baseline Data	INT (*n* = 10)	CON (*n* = 10)
Age (years)	12.8 ± 1.9	13.3 ± 0.17
Body mass (kg)	51.08 ± 8.36	54.33 ± 7.55
BMI (kg/m^2^)	18.02 ± 1.36	19.03 ± 1.34
Height (cm)	167.8 ± 9.9	168.85 ± 5.31
Body fat percentage (%)	5.37 ± 2.35	7.63 ± 4.93
Distance run per training (m)	6898.75 ± 1889.23	6720.42 ± 1384.18
Duration per training (min)	125.51 ± 19.83	125.81 ± 21.74
Low speed running (11–14 km/h) (m)	801.70 ± 133.9	729.06 ± 143.57
High speed running (>21 km/h) (m)	148.54 ± 41.46	133.46 ± 41.46
Maximum speed (km/h)	22.66 ± 1.13	23.17 ± 1.46
Training load (per minute)	6.65 ± 0.40	6.66 ± 0.71

**Table 2 ijerph-19-13186-t002:** Relative Lower Extremity Strength (MVC/BW).

Muscles	Intervention Group (*n* = 10)		Control Group (*n* = 10)	
	Pre- Intervention	Post- Intervention	Pre- Control	Post- Control
Knee Flexors	0.39 ± 0.05	0.42 ± 0.04 *,^&^	0.34 ± 0.09	0.33 ± 0.07
Knee Extensors	0.56 ± 0.10	0.50 ± 0.05 *	0.50 ± 0.11	0.50 ± 0.10

Note: Comparison before and after the intervention within the group: * = *p* < 0.05; Comparison between the control group and the intervention group during the same period: ^&^ = *p* < 0.05.

**Table 3 ijerph-19-13186-t003:** Changes in running ability.

		INT (*n* = 10)	CON (*n* = 10)
30 m-sprint (seconds)	pre-intervention	4.81 ± 0.26	5.11 ± 0.20
	post-intervention	4.39 ± 0.26 *,^&^	4.60 ± 0.17 *
5 × 25 shuttle run (seconds)	pre-intervention	36.01 ± 2.52	36.69 ± 2.16
	post-intervention	33.51 ± 1.55 *	35.25 ± 2.55

Note: Comparison before and after the intervention within the group: * *p* < 0.05. Comparison between the control group and the intervention group during the same period: ^&^ = *p* < 0.05.

## Data Availability

Not applicable.

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
