# Peer review of "The Effect of FIFA 11+ on the Isometric Strength and Running Ability of Young Soccer Players"

_ijerph, 2022, doi:10.3390/ijerph192013186_

Round 1

Reviewer 1 Report

Abstract - line 17 - change 'neuromuscular' to either 'muscular' or 'musculoskeletal'

Introduction - I feel this needs some further reference to previous research which highlights the importance of the physical attributes that you mention as important and targeted ie. strength, speed, agility etc.

Methods - I feel you need more detail in here around the actual exercises completed during the warm up time for the control group. This is important information to compare the exercises completed by each group. 

Methods - you mention about the plyometric portion of the FIFA11+ being adjustable based on the ability of the athletes. How was this controlled? How was the quantified? How was it determined what each subject did, and did this change throughout the experimental period?

Methods - line 114 - this is a typo error, needs correcting.

Methods - sections 2.5, 2.6, and 2.7 - these sections don't give enough detail around the actual tests and how they were completed. A higher level of detail is required. In my opinion, this section should allow me as the reader to complete the same test as you in the future if I wanted to. I couldn't do this in it's current format. 

Methods - lines 115 & 121 - I'm assuming here you are trying to reference a study with the '43' in these lines. If this is the case, reference 43 in the reference list doesn't correlate to a study that cites any work related to what you are talking about in lines 115 and 121. 

Methods - lines 115 & 121 - with the above in mind, on these lines you mention according to standard methods of a football physical fitness test. I'm not sure what this is. This needs either appropriate reference or further explanation. 

Methods - section 2.8 - this section requires more detail, particularly which software has been used for statistical analysis.

Table 1 - what is 'body fat rate'? is this a typo?

Results - I feel it is required to have more data presented for training load that just duration of sessions and total distance of sessions. A better understanding of training load may provide more context to the results of the tests completed in this study. Given the use of catapult GPS in this study, I feel it may be worthwhile to present variables such as high speed running, sprint distance, accelerations, decelerations and player load for example. 

Discussion - line 172/173 - you state that the results of this study show improvement in isokinetic and isometric strength. Firstly, how are you supporting an improvement in isokinetic strength as you haven't detailed any tests that measure this or presented any data around isokinetic strength? Secondly, for isometric strength, of the 4 conditions (knee flex INT, knee flex CON, knee ext INT, knee ext INT) only 1 of these improvement from pre to post intervention? Do you think this data actually supports this statement?

Discussion - in general, I feel the discussion requires a restructure and requires proof reading e.g. line 184. The flow of the writing needs to be improved from paragraph to paragraph. There also appears to be a paragraph (starting on line 182) which is 1 paragraph long. I'm not sure if this is a typo or not but the discussion generally needs re-working. 

There are format issues throughout, examples of this is line 121 where the '43' should be in brackets to show it is a reference. The use of the word 'football' should be 'soccer' for an international journal.  

Reviewer 2 Report

This manuscript focuses on a topic of sure interest to the readership of the International Journal of Environmental Research and Public Health. However, it has some issues. My main concerns are the following:

Flow of the content throughout the manuscript needs to be enhanced. Introduction and discussion could be clearer whilst the methodology needs some more explanations and adjustments. In the discussion authors go back and forth (e.g. with H/Q ratio and neuromuscular adaptation).

Other comments:

Line 36 and 37. A sentence describing injuries in recreationally active participants. What about others?

Line 39. Any numbers regarding injury incidence increment?

Line 50. ‘to function’ part can be excluded.

Line 54. Is ‘movement patterns immaturity’ an applicable term?

Line 57. You can delete ‘performance’ from the “speed and agility performance in football players”

Line 57. Is it current or available?

Line 57-59. Any details regarding research that only investigated the effectiveness of the intervention program on adults?

Line 63-65. Any details regarding research in adult participants and why it was expected that similar benefits would appear in adolescent participants?

Line 76. How was the randomization conducted?

Line 93. Any reference?

Line 114. Adjust numbers. 2.630.

Line 120. Protocol used?

Line 121. Method or protocol?; is 43 the number of cones?

Line 128. I suggest changing the analysis and using a 2-way repeated measures ANOVA. If not please explain why not.

Line 132. Remove space before ± in the parentheses

Line 171. Is 13 a reference?

Line 176. Is 15 a reference?

Line 180. Is 16 a reference?

Line 184. References? Brito et al.?

Line 201-203. What could be the underlying mechanism?

Line 214. Is Malone et al. number 38?

Line 339. Malone reference included full name. Please correct.

Line 347. Oxfeldt reference included full name. Please correct.

Round 2

Reviewer 1 Report

Thank you for making the suggested changes throughout the document. I understand this may have seemed like a lot of changes to make. 

I feel now the manuscript is in a much better place and just needs a proof read to ensure there are no grammatical errors in the document. I noticed a couple in the paragraph you added in response to point 2 of my original comments. I feel it would be best for the whole document to be checked in this sense.  

Reviewer 2 Report

Some changes have been made and further minor suggestions can be found below.

Line 81. Remove double “[6]”

Line 116. Remove “performance”

Line 145. Add space before [17]

Line 179. Add space before “30m”

Line 234. Table 1. Unify spaces before and after parenthesis throughout the whole table

Line 290. Remove “Error! Bookmark not defined”; modify [22][23][24] to [22-24]

Line 344. Remove “Error! Bookmark not defined”
